Diet of the endangered big-headed turtle Platysternon megacephalum

Sung Yik-Hei heisyh@gmail.com 1 2
Hau Billy C.H. 1
Karraker Nancy E. 1 3
1 School of Biological Sciences, The University of Hong Kong , Hong Kong SAR , China
2 Department of Biology, Hong Kong Baptist University , Hong Kong SAR , China
3 Department of Natural Resources Science, University of Rhode Island , Kingston , RI , United States of America
Kramer Donald
Electronic publication date: 2016 Dec 13
Publication date: 2016
Volume: 4
Electronic Location ID: e2784
Received 2016 Aug 5; Accepted 2016 Nov 11
Copyright: ©2016 Sung et al.
Copyright year: 2016
Copyright holder: Sung et al.
License: This is an open access article distributed under the terms of the Creative Commons Attribution License, which permits unrestricted use, distribution, reproduction and adaptation in any medium and for any purpose provided that it is properly attributed. For attribution, the original author(s), title, publication source (PeerJ) and either DOI or URL of the article must be cited.
License URL: https://creativecommons.org/licenses/by/4.0/

Keywords: Asian turtle crisis, China, Conservation, Functional ecology, Seed germination, Foraging ecology

Funding: The authors received no funding for this work.

==============================
Populations of the big-headed turtle Platysternon megacephalum are declining at unprecedented rates across most of its distribution in Southeast Asia owing to unsustainable harvest for pet, food, and Chinese medicine markets. Research on Asian freshwater turtles becomes more challenging as populations decline and basic ecological information is needed to inform conservation efforts. We examined fecal samples collected from P. megacephalum in five streams in Hong Kong to quantify the diet, and we compared the germination success of ingested and uningested seeds. Fruits, primarily of Machilus spp., were most frequently consumed, followed by insects, plant matter, crabs and mollusks. The niche breadth of adults was wider than that of juveniles. Diet composition differed between sites, which may be attributable to the history of illegal trapping at some sites, which reduced the proportion of larger and older individuals. Digestion of Machilus spp. fruits by P. megacephalum enhanced germination success of seeds by about 30%. However, most digested seeds are likely defecated in water in this highly aquatic species, which limits the potential benefit to dispersal. The results of our study can be used by conservation-related captive breeding programs to ensure a more optimal diet is provided to captive P. megacephalum.

Introduction

Populations of Asian turtles have been declining at rapid rates because of the insatiable demand for pet, food and traditional medicine markets (Cheung & Dudgeon, 2006). Over 80% of species are threatened and more than 50% are listed as endangered or critically endangered by the International Union for the Conservation of Nature. Populations of most Asian turtles have declined to such low levels that basic ecological studies are impossible for many species (Shen, Pike & Du, 2010).

The ecological roles of most Asian freshwater turtles remain unknown. Studies in North America showed that freshwater turtles can considerably influence ecosystem processes (Sterrett, Maerz & Katz, 2014) through movements of seeds and nutrients from aquatic to terrestrial habitats (Moll & Jansen, 1995) and potentially by enhancing seed germination (Braun & Brooks Jr, 1987). Understanding the ecological roles of and ecosystem services facilitated by endangered freshwater turtles can raise public awareness that is crucial for successful conservation (Mace, Norris & Fitter, 2012).

Populations of Platysternon megacephalum are declining at unprecedented rates across its distribution (Hendrie, 2000; Stuart & Timmins, 2000; Sung, Karraker & Hau, 2013; Tana et al., 2000; Wan et al., 2015), and thus it was recently proposed that its status be upgraded from Endangered to Critically Endangered (Horne, Poole & Walde, 2012). There is no evidence that harvesting is abating. Captive breeding of P. megacephalum has been attempted by various zoos and hobbyists, yet few instances of this species successfully breeding in captivity have occurred (Shelmidine, Murphy & Massarone, 2016; Sung, Hau & Karraker, 2014; Wei et al., 2016), which may be due to our limited knowledge about their natural history (Sung, Hau & Karraker, 2014). Only recently have researchers begun studying this species in the wild including distribution (Pipatsawasdikul, Voris & Thirakhupt, 2010), spatial ecology (Shen, Pike & Du, 2010; Sung, Hau & Karraker, 2015), growth (Sung et al., 2015) and reproduction (Sung, Hau & Karraker, 2014). The information gained from these studies will benefit conservation programs for the species, but much remains unknown. For example, information as basic as the diet of wild individuals is lacking. This turtle was long regarded as strictly carnivorous, suspected as feeding primarily on mollusks, crustaceans and fish, but that information was based solely on anecdotal observations (Bonin, Devaux & Dupre, 2006; Ernst & Barbour, 1989).

In order to develop conservation actions for endangered species, such as P. megacephalum, basic ecological information is needed. The objectives of this study were to (1) characterize the diet of wild P. megacephalum, (2) determine if this species exhibits ontogenetic changes in diet, (3) investigate variation in diet between sexes and ages, seasons and sites, and (4) investigate the effects of digestion on germination rate of Machilus seeds, which were the most frequently occurring diet item in fecal samples of P. megacephalum.

Materials & Methods

Study area

We conducted this study in five streams in the Hong Kong Special Administrative Region, China (22°09′–22°37′N, 113°50′–114°30′E). Elevations of the study sites ranged from 300–800 m above sea level, and riparian vegetation was mainly secondary forest dominated by Machilus spp. Among the five study streams, four were located in national parks and are accessible by the public and one is in a private, fenced and patrolled conservation area. We cannot disclose the exact locations of study sites to ensure the security of these populations; we refer to study sites as KF (private conservation area), MS, SH, TO and TN. All study sites were rocky streams characterized by fast flowing and clear water with shrublands or secondary forests in the riparian zone (Table 1). In Hong Kong, Platysternon megacephalum is protected under the Wild Animals Protection Ordinance Cap. 170, which prohibits any collection or use, but turtle populations in protected areas have been subjected to illegal harvesting (Sung, Karraker & Hau, 2013).

Table 1 Physical characteristics of the five study sites.

Riparian vegetation type, average and SD (in parentheses) of width and depth of microhabitats, and proportion of different substrate types (gravel (<2.0 cm), pebble (2.0–6.4 cm), cobble (6.4–25.6 cm) and boulder (>25.6 cm)) of the five study sites.

Site	Riparian vegetation type	Average width (cm)	Average depth (cm)	Percent of substrate	Past illegal trapping	
				Gravel	Pebble	Cobble	Boulder		
KF	Secondary forest	153.6 (144.7)	32.2 (45.0)	4.8 (14.1)	32.5 (32.8)	18.2 (25.3)	44.5 (73.6)	Absent	
MS	Shrubland and secondary forest	127.8 (113.7)	25.5 (22.9)	16.8 (17.2)	23.0 (23.6)	20.8 (20.0)	34.1 (33.6)	Present	
SH	Shrubland and secondary forest	111.5 (44.9)	16.7 (13.3)	31.3 (36.0)	46.3 (31.6)	11.3 (22.3)	11.3 (28.0)	Present	
TO	Secondary forest	100.6 (87.9)	23.4 (16.5)	17.6 (30.2)	41.4 (36.8)	20.5 (29.1)	20.5 (29.7)	Present	
TN	Secondary forest	168.2 (78.1)	30.3 (20.9)	0.9 (4.2)	45.2 (38.0)	21.3 (19.1)	32.6 (32.2)	Present	

Sample collection

Turtles were captured as part of a mark-recapture study (Sung, Karraker & Hau, 2013) carried out between September 2009 and June 2011, which included wet (April–September) and dry (October–March) seasons. We collected basic morphometric data on captured turtles, including straight-line carapace length (CL) using calipers, and body mass using a spring scale. We inserted passive implant transponder tags and used marginal scale notching following a system developed by Cagle (1939) to mark and identify turtle individuals. We sexed turtles by examining secondary sexual characteristics, including distance of cloaca from the edge of the plastron and thickness of the tail base above the cloaca, and all turtles smaller than 105 mm in CL were considered to be juveniles (Sung, Karraker & Hau, 2013). All procedures were approved by the Committee on the Use of Live Animals in Teaching and Research, the University of Hong Kong (CULATR 2249-10) and Agricultural Fisheries and Conservation Department of the Hong Kong Government (AF GR CON 09/51).

Because we were unsuccessful in attempts to use stomach flushing to obtain stomach contents as carried out by Legler (1977), we used fecal analysis to examine the diet following Demuth & Buhlmann (1997). This approach is minimally invasive and does not require sacrificing the animal. It allows identification of food items through the presence of undigested items such as invertebrate exoskeleton, bones, and seeds, but may underestimate presence or abundance of soft-bodied foods such as annelids and fruit. We collected fecal samples from captured turtles that were kept in plastic enclosures with approximately two cm of water for 20–24 h. We filtered the water and preserved the fecal samples in 70% ethanol.

We sorted the samples under a dissecting microscope (MZ8; Leica Microsystems, Wetzlar, Germany) and identified diet items to order or lower taxonomic levels if possible. Seeds of Machilus spp. were collected from fecal samples and were assessed for level of damage following digestion. Seeds that exhibited a spherical shape similar to undigested seeds were considered undamaged and were retained for the germination experiment. Disruption of the seed coat commonly occurs after ingestion by a vertebrate and may enhance germination (Traveset, 1998), but we considered a seed to be damaged if the seed endosperm was damaged. We documented proportions of seeds undamaged and damaged.

Seed germination test

To investigate the effects of gut passage on germination success and rate of Machilus seeds, we established three experimental treatments: undamaged digested seeds, undigested seeds with fruit pulp intact, and undigested seeds with fruit pulp removed. We collected undamaged digested seeds from fecal samples prior to preserving the rest of a sample in ethanol. On the same days that we collected digested seeds from fecal samples, we collected at least two fruits of Machilus spp. from the bottoms of study streams. We could not distinguish the fruits/seeds of different species of Machilus, but we only planted seeds collected in site KF between August and November, when only two species, Machilus breviflora and Machilus thunbergii, were fruiting. Seeds representing each treatment were planted in seed trays placed in a shaded area in a greenhouse. One seed was planted in each unit of a seed tray beneath 1 cm of potting soil. Seeds of all treatments were planted in four identical trays in a randomized complete block design, with each tray containing eight to sixteen replicates of each treatment. Seeds were watered approximately three times per week, depending upon ambient temperature and drying of the potting soil. Seeds trays were checked at least three times per week for six months and germination was documented. Seeds that did not germinate within six months after planting were regarded as unviable.

Data analysis

To avoid pseudoreplication, we randomly selected one fecal sample for individuals from which multiple samples had been collected for analysis. We calculated the frequency of occurrence of each diet item as percent of individuals that contained a given diet item (Bowen, 1983). We calculated niche breadth of female, male and juvenile turtles using the Shannon index: H′= ∑i=1npi lnpi where pi is the frequency of occurrence of diet item i in a particular age and sex group, and season (Magurran, 1988; Sargeant, 2007). We standardized H′ on a scale of 0 to 1 using an evenness index: J′ = H′(lnn)−1 where n is the number of diet categories (Pielou, 1969; Sargeant, 2007).

Ontogenetic changes in diet may represent a continuous transition, and an ontogenetic shift in diet may not be easily detected by comparison of sex and age groups. Therefore, we conducted logistic regression analyses with occurrence of seeds and animals as response variables, and carapace length of turtles as predictor variables. We analyzed occurrence of fruits instead of all plant matter as plant matter excluding fruits, mainly unidentifiable vegetative matter, frequently occurred (38–50%) but in trace amount in fecal samples, which may indicate incidental ingestion when consuming other diet items (Demuth & Buhlmann, 1997). We included all data including empty fecal samples to eliminate the potential effects of body sizes on the occurrence of emply stomachs. We performed the analysis in R (R Development Core Team, 2014).

We also conducted multivariate analysis to compare diet composition among seasons, sites, and age and sex groups based on presence of diet items with non-metric multidimensional scaling and analysis of similarity using Bray–Curtis similarity index. We used similarity percentage procedure to determine the contribution by each diet item to the differences among seasons, sites, and age and sex groups. We conducted analyses using PRIMER 6.0 (Clarke & Warwick, 2001).

We compared germination success of seeds collected from fecal samples and control seeds with and without pulp using a generalized linear mixed model with a binomial error variance (Zuur et al., 2009). We included seed tray as a random factor. Seeds from fecal samples were regarded as the reference category in the Wald Z test. We performed the analysis in R (R Development Core Team, 2014) using glmer in the lme4 package (Bates, 2010).

Results

We collected 141 fecal samples, of which 89 contained at least one item, from 61 individual turtles (31 females, eight juveniles and 22 males). We identified 356 diet items belonging to 11 categories (Table 2). Diet items most frequently recovered from fecal samples were fruits, insects and mollusks. All fruits recovered belonged to the genus Machilus, except one sample that contained seeds of Turpinia arguta. Identifiable remains of insects consisted of terrestrial adults and larvae belonging to seven orders (Coleoptera, Homoptera, Hymenoptera, Isoptera, Lepidoptera, Mantodea, Orthoptera) and aquatic larvae belonging to four orders (Diptera, Ephemeroptera, Odonata and Tricoptera). All mollusks found were Sulcospira hainanensis. We also recovered parts of other animals including frog bones, bird feathers, fish bones, rodent bones and freshwater crab shells.

Table 2 Frequency of occurrence of food items in fecal samples of Platysternon megacephalum.

Frequency of occurrence of food items in the diet of juveniles (J), females (F), and males (M) of Platysternon megacephalum in five streams in Hong Kong between 2009 and 2011.

Diet item	Wet season	Dry season	All	
		J	F	M	J	F	M	J	F	M	
Plant	Fruit	66.7	68.0	75.0	50.0	80.0	85.7	62.5	70.0	78.3	
	Other plant matter	50.0	40.0	37.5	50.0	40.0	42.9	50.0	40.0	39.1	
Animal	Mammal	0	0	0	0	0	14.3	0	0	4.3	
	Bird	0	0	6.3	0	0	0	0	0	4.3	
	Frog	0	4.0	0	0	0	0	0	3.3	0	
	Lizard	0	4.0	0	0	0	0	0	3.3	0	
	Crab	0	24.0	25.0	0	60.0	28.6	0	30.0	26.1	
	Fish	0	4.0	12.5	0	0	0	0	3.3	8.7	
	Mollusk	16.7	20.0	18.8	50.0	40.0	42.9	25.0	23.3	26.1	
	Insect	66.7	68.0	62.5	0	40.0	0	50.0	73.3	43.5	
Unidentified matter	0	4	0	50	0	0	12.5	3.3	0	
Sample size	6	25	16	2	5	7	8	30	23	
Shannon index	1.26	1.86	2.12	1.04	1.58	1.49	1.33	1.88	2.11	
Evenness index	0.55	0.81	0.92	0.45	0.69	0.65	0.58	0.82	0.92	

Based on the Shannon and evenness Indices, niche breadth in the wet season was broader than that in the dry season (Table 2), and niche breadth of adult turtles was wider than that of juvenile turtles. Niche breadth in males was wider in the wet season but narrower in the dry season than in females.

There was a significant positive relationship between carapace length of turtles and occurrence of fruits (Z = 2.12, P = 0.034), but the relationship was not significant between carapace length of turtles and occurrence of animals (Z = 1.53, P = 0.127).

Diet composition was similar between seasons (R = 0.074, P = 0.115), and sexes and ages (R =  − 0.009, P = 0.570), but differed among sites (R = 0.344, P < 0.001; Fig. 1). In pairwise comparisons, diet composition of turtles in KF differed from that of other sites (P < 0.020), whereas the diet of turtles in MS, TN and TO was similar (P > 0.332). Diet of turtles in SH was different from that in MS (P = 0.024) but similar to diet of turtles in TN (P = 0.332) and TO (P = 0.075). Fruits, insects, crabs and other plant matter contributed the most to the dissimilarity between sites (Table 3). Frequency of occurrence of fruits was the highest in KF and that of insects was the lowest, whereas frequencies of occurrence of fruits and crabs were the lowest in SH (Fig. 2).

Figure 1 Composition of food items in different seasons, by different age and sex groups and in different sites.

Two-dimensional non-metric multidimensional scaling representing Bray–Curtis distances among composition of food items consumed by Platysternon megacephalum (A) in different seasons, (B) by different age and sex groups, and (C) in different study sites in Hong Kong between 2009 and 2011.

Table 3 Dissimilarity percentages (lower diagonal) and the two diet items that contributed the most to the dissimilarity in diet between sites (upper diagonal).

Pairwise comparison table showing dissimilarity percentages (below diagonal) and the two diet items that contributed the most to the dissimilarity in diet of 61 Platysternon megacephalum between five study sites (above diagonal; contributing percentage in parenthesis) in Hong Kong between 2009 and 2011.

Site	Sample size	Mean carapace length (±SD)	KF	MS	SH	TN	TO	
KF	32	130.2 (±33.0)		Fruit (30)	Insect (22)	Fruit (26)	Fruit (28)	
	Crab (26)	Fruit (20)	Plant (20)	Plant (19)	
MS	3	106.6 (±6.8)	79		Crab (31)	Crab (28)	Insect (35)	
	Fruit (18)	Insect (26)	Plant (23)	
SH	13	108.7 (±23.7)	54	79		Plant (21)	Crab (20)	
	Fruit (20)	Plant (20)	
TN	7	102.3 (±10.6)	62	53	64		Crab (23)	
	Plant (20)	
TO	6	114.1 (±17.0)	69	42	66	48		

Figure 2 Frequency of occurrence of the five most dominant diet items in the five study sites.

Frequency of occurrence of the diet items most frequently recovered from fecal samples from 61 Platysternon megacephalum in five study sites in Hong Kong between 2009 and 2011.

Of seeds consumed by turtles, 64% were damaged, either by mastication or the digestion process. Of intact seeds that were planted, 37.5% (12/32) germinated, compared with 3.6% (2/56) of control seeds with pulp removed (Z =  − 3.45, P < 0.001) and 2.9% (1/35) with pulp intact (Z =  − 2.80, P < 0.005).

Discussion

This is the first study to quantify the diet of the endangered Platysternon megacephalum in the wild. P. megacephalum have long been regarded as carnivorous (Bonin, Devaux & Dupre, 2006), but we found that fruits were frequently consumed, and we believe that this fruit is consumed within the stream channel. Fruits occurred in at least 62.5% of fecal samples of females, juveniles and males. In Hong Kong, complete deforestation occurred before the Second World War (Corlett, 1999), and trees in the genus Machilus have become the dominant species in secondary forests (Zhuang, 1997). At least four species of Machilus occur in the riparian habitats of the study streams. Fruiting of these Machilus species spans from March to August and October to December (AFCD, 2008), and the steep banks of hillstreams inhabited by P. megacephalum serve to channel large quantities of Machilus fruits downslope and into the streams. Thus, fallen fruits provide a constant food supply to P. megacephalum through most of the year. Fruits of Turpinia arguta were also consumed, and consumption of Ficus fruits has also been reported by illegal hunters in Hainan, China (YH Sung, 2013, unpublished data). Given the broad distribution of P. megacephalum in Asia, they likely consume a higher diversity of fruits than observed in this study.

P. megacephalum exhibits ontogenetic shift in diet, becoming increasingly frugivorous with increases in body size. Shifts in diet from largely carnivorous to largely herbivorous have been documented in a number of freshwater turtles (Chen & Lue, 1998; Parmenter & Avery, 1990; Spencer, Thompson & Hume, 1998). However, a high proportion of adult P. megacephalum consumed a diversity of animals, including larger prey, such as frogs, fish, and crabs, upon which smaller juveniles are incapable of predating and thus contributing to the narrower niche breadth of juveniles. It was surprising that fruits of Machilus occurred in 62.5% of fecal samples of juveniles, including the smallest juveniles with carapace length of 48 mm, indicating that fruits may be an important diet item across turtles of all sizes. The diet of juveniles may require further investigation because of the small sample size.

Diet composition of P. megacephalum differed between study streams, and this may be associated with demographic differences among sites, which have been shaped by a history of illegal trapping. Illegal trapping has depleted populations, resulting in lower densities of large adults and smaller average body sizes in all study streams except in KF, the private conservation area (Sung, Karraker & Hau, 2013). The KF population, which exhibits the sex and age structure of a healthy population (Sung, Karraker & Hau, 2013), consumed fruits more frequently and animals less often, compared to other populations (Fig. 2). It should be noted, however, that we do not have data on availability of diet items, so we cannot disregard this explanation for our results.

The importance of animals in the diet of P. megacephalum may be more pronounced than it appeared in this study. We found that the most dominant animal prey items were crabs, mollusks, and beetles, which all have hard exoskeletons. Remains of small or soft-bodied animals, such as earthworms, were underrepresented in fecal analysis. Thus, the relative importance of fruits may be overestimated as has been suggested in some other omnivorous turtles (Caputo & Vogt, 2008; Platt et al., 2016). We observed three predation events by P. megacephalum in streams, including predation of an Anderson’s stream snake Opisthotropis andersoni, an adult dung beetle, and a moth larva (YH Sung, pers. obs., 2015 (snake), 2010 (dung beetle), 2010 (moth larva)). Stable isotope analysis will be complementary to this study and useful to further determine the relative importance of different diet items and elucidate the species’ trophic position in the ecosystem (Bearhop et al., 2004).

The occurrence of bird feathers and rodent bones in fecal samples suggested that P. megacephalum may be opportunistic scavengers, but it is not clear how important scavenging is to the diet of these turtles. Other freshwater turtles, such as Macrochelys temminckii, have been reported to scavenge on mammals (Elsey, 2006), and it is likely that most carnivorous and omnivorous species opportunistically scavenge. As densities of P. megacephalum can be relatively high in protected populations (Sung, Karraker & Hau, 2013) and other large aquatic vertebrates do not occur in these systems, P. megacephalum may play an important role as scavengers and thus in nutrient cycling (Sterrett, Maerz & Katz, 2014) within these aquatic ecosystems and occasionally at the land-water interface.

The germination success of Machilus seeds ingested by P. megacephalum was about 30% higher than that of seeds had not been ingested. Although 65% of seeds in fecal samples were damaged, enhanced germination success following ingestion by this turtle compared with the very low germination success (<4%) of uningested seeds probably outweighs the damage to some seeds. Enhanced germination success of seeds ingested by turtles has been documented in other species (Braun & Brooks Jr, 1987; Cobo & Andreu, 1988; Rust & Roth, 1981), but most studies have focused on tortoises that both ingest and defecate seeds in terrestrial habitats. To our knowledge, only two studies (Kimmons & Moll, 2010; Moll & Jansen, 1995) have examined the effects of ingestion by aquatic turtles on seed germination. Given life histories of the focal species of those studies, each would be capable of ingesting seeds in an aquatic habitat and defecating them in a terrestrial habitat, thereby transporting seeds from aquatic habitats where germination is unlikely to terrestrial habitats where it is possible. However, of three species examined, ingestion of plant seeds by Rhinoclemmys funerea in Costa Rica (Moll & Jansen, 1995), and Trachemys scripta and Chelydra serpentina (Kimmons & Moll, 2010) in the US did not enhance germination. Ingestion of seeds by P. megacephalum increases germination success, but this is only beneficial if the turtle periodically leaves the aquatic habitat.

In previous research on this species’ spatial ecology (Sung, Hau & Karraker, 2015a), we found that individuals are highly aquatic and make few movements away from the stream. However, we believe that we probably underestimated terrestrial movements because turtles were occasionally observed in terrestrial habitats during/after extreme storm events in the monsoonal wet season, when these high velocity, torrential streams are far too dangerous to be visited by researchers. For example, on 15 days between May and September 2010, there were rainstorms of a severity level (Hong Kong Observatory, 2016) that would likely have driven turtles out of streams and have made the streams too dangerous for researchers. In addition to leaving streams during major storm events, females must also leave streams to nest. Flooding in streams may assist digested seeds in returning to riparian forest floor from water. In mainland China, translocated turtles purchased from markets spent about 7% or their time on land (Shen, Pike & Du, 2010), but it is not known how their habitat use differs from that of turtles in their original streams. Although probably contributing to seed dispersal, the role of P. megacephalum may be less important than that of other groups, such as frugivorous birds (Corlett, 2011).

Conclusions

Rapid population declines and low densities of Asian freshwater turtles have limited opportunities for ecological study. Although P. megacephalum have disappeared across much of China (Lau & Shi, 2000; Shi et al., 2007), populations remain in Hong Kong. We found that P. megacephalum are omnivorous and may facilitate important ecological processes, including cycling of plant and animal matter in the aquatic ecosystem and potentially aid in seed germination. We recommend that future research includes stable isotope analyses to identify the roles played by this endangered species in the food chain while populations remain. Such information on this species and other freshwater turtle species in Asia may lead to greater awareness about the need for conservation. Captive breeding program managers may refer to the results of this study to provide a more optimal diet, including the provision of fruits, to captive P. megacephalum.

Supplemental Information

Supplemental Information 1 Raw data of diet analysis

Click here for additional data file.

Supplemental Information 2 Raw data of seed germination test

Click here for additional data file.

We are grateful to the staff of Kadoorie Farm and Botanic Garden, and particularly A. Brown, G. Ades, P. Crow, R. Kendrick, A. Grioni, and M. Lau, for supporting this project in many ways and for their dedication to the conservation of Asian turtles. Thanks to L. Ng, M. Lo, and L. Wong for technical and logistical support and numerous biologists for their assistance in the field.

Additional Information and Declarations

Competing Interests

Author Contributions

Animal Ethics

Field Study Permissions

Data Availability

The authors declare there are no competing interests.

Yik-Hei Sung conceived and designed the experiments, performed the experiments, analyzed the data, wrote the paper, prepared figures and/or tables, reviewed drafts of the paper.

Billy C.H. Hau and Nancy E. Karraker conceived and designed the experiments, contributed reagents/materials/analysis tools, reviewed drafts of the paper.

The following information was supplied relating to ethical approvals (i.e., approving body and any reference numbers):

Committee on the Use of Live Animals in Teaching and Research, the University of Hong Kong (CULATR 2249-10).

The following information was supplied relating to field study approvals (i.e., approving body and any reference numbers):

Agricultural Fisheries and Conservation Department of the Hong Kong Government (AF GR CON 09/51).

The following information was supplied regarding data availability:

The raw data has been supplied as Supplemental Files.

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
