# Peer review of "Diet of the endangered big-headed turtle Platysternon megacephalum"

_PeerJ, doi:10.7717/peerj.2784_

## Round 0.1 · original submission · Major Revisions

This was a descriptive examination of the diet of endangered big-headed turtles based on identification of taxa and their volume in fecal samples from which a number of dietary measures were calculated, including the frequency, abundance, volume and an Index of Relative Importance for each diet category, Levins' Niche Breadth Index, species richness and the Shannon Index of Diversity. Comparisons were made between males, females and juveniles in the wet and dry season using GLMMs for abundance, volume and species richness and non-metric multidimensional scaling and Bray-Curtis similarity index for diet composition. In addition, the authors compared germination of seeds collected from feces with germination of seeds collected from the study streams. Turtles consumed a variety of plant and animal material. Abundance of diet items was higher in the dry season, volume was higher in the most protected site, and richness did not vary among seasons, sexes and sites. Niche breadth of juveniles was greater than that of adults and Shannon diversity was higher in the wet season. Diet composition differed among sites, apparently due to the importance of seeds and mollusks, but was similar among seasons, sexes and age classes. Although many seeds were damaged during ingestion and digestion, surviving seeds had a higher germination rate than seeds removed from the streams.

Reviewer 1 expressed support for the publication of this manuscript and recommended minor revision to address some concerns about the statistics. The reviewer was unable to submit his review through the web site because of his travels so sent a review and annotated manuscript directly to the journal. Although his comments in the overview were rather brief, he/she provided a considerable number of useful comments on substance and style in the annotated manuscript. These include the need for more detailed description of the general characteristics of the study streams, questions concerning how multiple samples from the same individuals were addressed, the design of the germination experiment, and several aspects of the discussion that need elaboration.

Reviewer 2 was also positive about the contribution of the manuscript. However, he/she expressed concerns that inclusion of several unnecessary analyses contributed little to the conclusions and obscured the main message, that methods for calculating volume and abundance of dietary components were inadequately described and were inappropriate for fecal analysis, and that the authors did not explain how they dealt with multiple samples per individual. This reviewer also felt that a more critical approach to the interpretation of seed germination data was needed and noted several other points where the manuscript needed clarification or correction of grammar. This reviewer recommended rejection of the manuscript, but his/her comments corresponded more closely to a recommendation of major revision. It is possible that they interpreted 'rejection' as 'reject and resubmit' as is commonplace in other journals.

My own reading of the manuscript led me to agree with the reviewers that this is a valid contribution to understanding the diet of a poorly-studied species and that their suggestions would make substantial improvements in the presentation. In addition, I had a number of concerns about the completeness of the Methods, the link between conclusions and results, and the overall organization of the manuscript. I therefore recommend major revisions.

I have provided a detailed review below and added my own comments to the annotated pdf submitted reviewer 1. In your response to the reviews, you should list the suggestions made by reviewer 1 on the pdf and respond to them. My substantive suggestions are all listed below; you may respond to them as you would to any reviewer (modify the manuscript appropriately if you agree or provide a clear, logical rebuttal if you disagree). You do not need to reply to my grammatical suggestions on the manuscript unless you do not intend to follow them.

Editor's Review

General concerns

With regard to the unnecessary and inappropriate analyses, I agree that measures of volume, abundance and related Index of Relative Importance of undigested material in fecal samples provide a strongly biased measure of importance in the diet, one that could mislead an uncritical reader. You need to either remove or much more fully justify these measures.

For the analysis, it is critical that all terms and all calculations be carefully defined, including units. For example, frequency of occurrence (L131), abundance (L140), and species richness (L140) were not explicitly defined. (Note also that you did not identify items to species level, so the normal definition of species richness does not apply.) Levin's niche breadth (L135) lacks a reference and it is unclear how a formula relating to only a single dietary type can measure niche breadth; is a summation sign missing? The formula for the Shannon Index appears to have substituted In for ln. When correcting formulas, please confirm that the calculations were all done correctly rather than following the erroneous formulations.

As a reviewer indicated, when the data are not normally distributed, medians (and quartiles) are more appropriate measures of central tendency (and spread) of the data. Check your distributions for normality and see whether you need to revise your descriptive statistics.

The organization of the manuscript needs to have better integration. Your present statement of your objective (L78-79) is much too vague ('quantify diet', 'investigate ecological roles'). You should be much more specific so that you can relate each of your analyses back to a specific objective. This applies to both the Methods and to the Results sections. Furthermore, your Discussion should follow the order of your objectives to facilitate the readers' understanding and should address each of the objectives and results. At present, there is little or no discussion of several of the results.

The Results (L167ff) makes the error of confounding 'not significantly different' with being the same or similar. It is possible that lack of a significant difference is due to high variability or small sample size rather than to similarity of the values. Furthermore, data showing the central tendency and distribution of the data for sites is not shown. Also, Table 2 does not provide units for the measures.

More specific concerns

L34 ff. The Abstract should be revised to put less emphasis on background and general statements that are not conclusions or direct implications of your work (currently more than half the abstract) and more emphasis on the original findings of this study (currently many measures are ignored). You must specify your method and make sure you don't imply that your variables measure consumption rather than fecal composition. I think it is an exaggeration to state that you studied the ecological roles of the species when all you did was study germination of seeds that passed through the gut with potential implications for one ecological role. I don't think your discussion of the germination effect allows you to claim 'an important role in cross-ecosystem resource subsidies', even as a suggestion. The final sentence should focus on implications of your work. Using ecological roles to gather support for conservation is a pre-condition not an implication of your work as you present no information about increased support. Recommending further studies is too vague and general and would have been obvious before your study. What specific questions related to your work are outstanding? (These improvements in the Abstract should obviously link to related points developed more strongly in the Discussion.)

L67-68. It is not clear what you mean by little success after stating that captive breeding has been carried out. Do you mean that it has been attempted with little success?

L83ff. You imply that the study sites were separate streams rather than different sections of the same stream(s). Without revealing too much, can you say something more about the habitat, such as indicators of the size and type of stream, water temperature, substrate, aquatic vegetation and other relevant flora and fauna? It seems that such information may be important for any future comparative studies.

L96. How were turtles marked?

L104ff. Start this paragraph by an explicit statement and providing references to other studies that used this approach and addressing the limitations and values of this approach. Something along these lines will introduce your method more fully to readers: 'Because we were unsuccessful in attempts to use stomach flushing as carried out by (ref), we used faecal analysis to examine the diet following (ref). This approach is minimally invasive and does not require sacrificing the animal. It allows identification of food items through the presence of undigested items such as invertebrate exoskeleton, bones, and seeds, but may miss soft-bodied foods such as annelids and fruit.'

L111 and Fig. 1. What is the justification for the lower taxonomic separation of insect remains as compared to other groups? Does this create a bias by producing multiple small categories when insects might be frequently consumed? Wouldn't it be more appropriate to have a broad category for insects as you do for mollusks and crustaceans? In the written results, you might indicate in more detail whether the prey were aquatic larvae and nymphs or adults and whether they were from terrestrial or aquatic taxa, if you can say.

L112. What equipment did you use that allowed you to measure displacement to a precision of 0.1 ml? Do you need a reference for your approach?

L114. Briefly describe how you defined a damaged seed?

L117. The seed germination study is presented too sparsely for the reader to judge its validity. There is no information about the sample size, the species distribution in each treatment, or the distribution in time and space of the source of the seeds for each treatment. This leaves lots of room for invalid conclusions, for example if turtles consumed different species from those collected in the streams or if the seeds in turtle feces were from different populations or seasons of the seeds collected in streams, or if the seeds in streams had been in the water longer than the seeds in feces. Also, you did not specify the environment in which the germinating seeds were maintained (sun, shade, indoors, outdoors). What was the total number of replicates?

L129ff. I think that you should reconsider the order in which the data analysis methods are presented. To me, it makes more sense to explain measures that are made on each individual before explaining the measures based on the aggregated samples. For example, you could describe the calculation of frequency (and volume and abundance, if used), then the methods to examine how you examined the effect of sex/age class, season and site on each of these measures, then breadth, diversity, richness (and IRI, if used). Also, if individual measures vary by sex or season, would it make sense to calculate the aggregated measures for each grouping separately as well as combined?

L173. It is not clear to me how niche breadth would be wider in juveniles considering how many items consumed by adults were not found in juveniles. Could this be an artifact of sample size or the multiple categories of insects as compared to other taxonomic groups?

L181. You need to explain more fully how Table 4 supports the above statements.

L192. Because your analysis is based on fecal samples including largely undigestible seeds, the importance of fruit in the diet is an indirect inference. You need to make a logical argument before drawing the conclusion that fruit is a major component of the diet.

L225. What is the logic of assuming that turtles move into terrestrial habitats (as opposed to quieter waters near the sides of the flooded stream, as fishes do, for example)?

L253. You need more explicit discussion of the link between herbivory and niche breadth, including more specific reference to the supporting information in the Results.

L256. Check your results: you explicitly stated that abundance did not differ between populations, and diversity was an aggregate measure that was not compared between populations. If there is a difference in the protected population, you need to explain how a higher density/lower exploitation population could lead to higher abundance and diversity in a population. Is there any literature supporting this pattern?

L259. Is there any literature on related species to which you can compare your results, qualitatively and/or quantitatively? How novel is the potential importance of frugivory for a turtle?

L264-265. You have overstated your findings here. You did not even investigate or discuss the role of this species in the food chain; consider what sort of evidence you would need to support this statement. You did not make the case that the germination effect was an important ecological process. In fact, you and the reviewers both indicated that it is more likely to be very minor. In suggestions for future studies, be more explicit about what sort of studies you feel would be helpful, drawing specifically from what you found in your study. Similarly, explain implications for captive breeding.

Figure 1. Do the nMDS plots require axis labels?

Reviewer 1 ·

Basic reporting

.

Experimental design

.

Validity of the findings

.

Additional comments

In essence, this is solid basic ecology/life history manuscript. What makes it important is that it is on a species that is under serious threat and there is very little data of this sort to help conserve and manage populations throughout Asia. Asian turtles are all but wiped out and are under serious threat. More papers like this are required. So, yes this paper should be accepted.

I think the authors need to readjust some aspects of the manuscript. they need to pay attention to some of the data analysis where they might have multiple samples from the same individual. This might just need more explanation, but I suspect a slight re-analysis might be required. i doubt it will affect the interpretation very much. Also the stress values of the nMDS are quite high..

I have also indicated parts of the discussion that need expanding.

Annotated reviews are not available for download in order to protect the identity of reviewers who chose to remain anonymous.

Reviewer 2 ·

Basic reporting

The background and importance of this research was well stated and justified. The authors are correct that basic life history data is missing for many imperiled species, and collecting those data are vital to conservation efforts. So I applaud their efforts to do so.

Most of this paper was well written, and the introduction and discussion were easy to follow. However, the methods and results were convoluted and difficult to follow. They were cluttered with many unnecessary analyses that were poorly explained. I think that this paper would be much easier to follow and more useful it was limited to the core results/comparisons that are likely to be useful for conservation efforts and understanding the ecology of this species (richness, diversity, and niche breadth).

The figure is appropriate, but I think that it would be very useful to include a few figures comparing the percent of dietary items found in each major category for different turtle groups. For example, the authors could have a bar graph comparing the wet season and dry season where the first pair of bars is the percent of dietary items that were plants, the second pair is the percent that were amphibians, etc. They could also do this for the sexes/juveniles, and sites. I think that this would be a very nice visualization of the data that would help readers to see the big picture.

The data files were provided, but they need to either be modified or include a key. Right now, diet_ID, season, site, and sex are all coded as numbers, but it is unclear what the numbers refer to, which makes it impossible to analyze the data set (e.g., is the dry season 1 or 2? what is diet item #72? etc).

Experimental design

I have serious concerns about the analysis methods that were used.

Use of volume: Multiple comparisons were made using the volume of the food items, but I am highly skeptical of the appropriateness of that method for these data. Typically, only the hardest parts of the dietary intake will remain intact after digestion (e.g., seeds, exoskeletons, bones, etc.), and you can’t assume that the volume of those items accurately represent the volume of the food items that were initially ingested (i.e., a seed has a vastly different volume than the original fruit). To put this another way, with faecal samples, you aren’t looking at what the turtle ingested. Rather, you are looking at what the turtle was unable to digest, and you are using those data to infer information about what was originally ingested. As a result, volume can give extremely misleading results.

For example, for females, the frequency of occurrence for Machilus was 22.6 and for crabs it was 10.5, yet the percent volume of Machilus was 80.5 and the percent volume for crabs was only 4.6. This makes it seem like Machilus is the most important component of females diets, while crabs play a relatively small role. However, that may not be correct, because a crab’s exoskeleton (which would have remained undigested and been used to measure volume) only represents a small portion of the volume of an actual crab. Conversely, some Machilus species have very large seeds, so the seeds (which would have remained undigested and been used to measure volume) make up a substantial portion of the volume of the original fruit (often, the seeds are even larger than the portion of the fruit that turtles can digest). Thus, using volume could severely overestimate the importance of Machilus in turtles’ diets, while underestimating the importance of crabs (and other animals). In short, the volume of undigested matter is not a good proxy for the volume of digested matter when you are dealing with such disparate food items.

I don’t want this to sound too disparaging, because faecal data are very useful for many purposes (such as the diversity metrics that the authors used elsewhere), but I don’t think that they can be used for volumetric comparisons.


Multiple samples per individual: The authors stated that multiple samples were collected for some individuals, but that would create pseudoreplicaiton, and it is not clear how that was dealt with. Looking at the data that was provided, I suspect that the authors took an average for each individual before running the statistics. That is an appropriate way to deal with this problem, so if the authors did that, then they simply need to state what they did. If, however, the authors did not do that, then they need to either re-run the stats after taking those averages, or randomly select one sample per individual and rerun the stats using those samples. Regardless of which solution is used, it needs to be applied to all of the diversity calculations and statistical comparisons.


Mean volume and abundance: The authors both report and conduct GLMMs on the mean abundance and volume of prey items, but it is not clear to me what those measurements are or why they are important. For mean volume, for example, my guess is that for each turtle this is the mean volume of all prey items, but that is not clear. Further, if my understanding is correct, then I don’t know what the significance of that is. Why does it matter that one site had a greater mean volume or a greater mean abundance? I’m just not sure what that actually tells us. It seems like richness, diversity, and niche breadth are the metrics that actually matter, and including these means just bogs the paper down and makes it hard to follow without actually adding anything useful. Finally, looking at the data sheets provided by the authors, these data are highly non-normal. So if the authors think that it is important to include these metrics, the medians should be reported rather than the means.


Lines 117–119: The authors stated that some seeds were removed from the samples before preserving the samples for later analysis, but it is not clear how they accounted for this when calculating the composition of the turtles’ diets.


Lines 142–145: I think that site should be a random factor in these models. Presumably these are not the only 5 sites where these turtles are found, and the authors would obviously like to generalize beyond this study.

Validity of the findings

In addition to the methodological problems identified above, I am skeptical about the importance of this species for seed dispersal, and the authors seem to be aware of the problems with suggesting that a largely aquatic species is an important seed disperser, but they try to make that argument anyway. For example, they argue that the turtles may move on land during heavy rainfall events. Although that may be true, it is speculative and occasional events like that would have a minimal effect on the community. Additionally, their experiment failed to test an extremely important group: seeds that fall on the ground. They compared the seeds that turtles ingested to seeds from the bottom of the river, but water-logged seeds may have a vastly different germination rate than seeds that were deposited on land. Further, the germination rates of seeds from the river seems quite irrelevant because those seeds presumably would never germinate in the wild because they are at the bottom of a stream. So, to establish that this turtle is a beneficial seed disperser, the comparison needs to include seeds on land that have the actual potential to germinate, but that group is lacking from this experiment.

I also have two concerns about the data presented in table 3. First, I don’t understand the output in these tables with regards to study site. They are supposed to be the output from a GLMM with season, sex, site, and individual as the factors, but that should produce a single P value for each of those factors, so why is there a separate P value for each site? Are these post hoc P values rather than GLMM P values? If so, what were the GLMM values? Right now it looks like post hoc tests were performed even though the GLMM probably wasn’t significant. Please clarify this (also see previous comments on mean abundance and volume).

Second, I find it shocking that species richness was not significant for season given the data shown in table 2. Based on that table, it looks like there is an enormous difference. I think that this disparity merits an explanation/discussion.

Additional comments

Dear authors, this is a potentially interesting and important study. I know from personal experience that this type of project is a massive undertaking, and I think that the data that you collected are timely and extremely useful for conservation efforts. However, I do have serious concerns about some of the methods, and I cannot recommend publishing it at the moment. Nevertheless, I think that by reducing this paper to the core, reliable results (richness, diversity, and niche breadth) and avoiding the use of inappropriate metrics like volume, you can produce a very useful and solid paper with the data that you have collect. I do hope that you choose to do this, because these data are valuable and I would like to see them published. In addition to the major comments provided above, I have included a few line-specific comments below. I hope that these will be helpful for you.

Lines 54–56: This is not a complete sentence. Change to “…such low levels that basic ecological studies are impossible for many species”

Lines 65–66: Move “recently” forward in the sentence so that it reads “it was recently proposed.” Also, remove the extra space between “from” and “Endangered.”

Lines 104–109: I agree that the use of faecal samples is a valid method for examining diets in turtles (especially when stomach flushing has failed); however, many researchers insist on using stomach flushing for turtles. Therefore, I recommend citing other papers that used faecal samples in order to establish the validity of this method. The following papers all studied turtle diets via collection methods that are similar to yours and you should consider citing them.

McKnight DT, Jones AC, Ligon DB. 2015. The omnivorous diet of the western chicken turtle (Deirochelys reticularia miaria). Copeia 103:322–328.

East MB, Ligon DB. 2013. Comparison of diet among reintroduced and wild juvenile alligator snapping turtles (Macrochelys temminckii) and adult female Ouachita
map turtles (Graptemys ouachitensis). Southwestern Naturalist 58:450–458.

Demuth JP, Buhlmann KA. 1997. Diet of the turtle Deirochelys reticularia on the Savannah River site, South Carolina. Journal of Herpetology 31:450–453.

Lines 130–132: The spacing between these lines is different than the spacing used elsewhere.

Lines 132–133: You collected gut contents, not stomach contents

Lines 153–157: Why not just use an ANOVA? The method used here isn’t necessarily wrong, but it seems unnecessarily complicated when this is precisely the type of experiment that ANOVAs were designed to analyse. Most readers are more familiar with ANOVAs and have an easier time understanding their output, and I personally think that it is best to go with the method that is easiest for the majority of readers to understand (assuming that the assumptions of ANOVA are met by your data).

Lines 205–206: Use precise numbers rather than “about.”

Line 347: Delete blank line

Line 404: Delete blank line

Table 2: See previous comments about mean abundance/volume. Also, the species richness listed here is a mean value, but I think that it would also be very interesting to see the total species richness (i.e., pool all individuals within a category and look at the entire richness). Right now, what you have shown is that the richness of species in a turtle’s gut at a given time is greater in the wet season than in the dry season, but it would also be interesting to know if the richness of all prey items eaten during the seasons differ.

---

## Round 0.2 · Minor Revisions

This manuscript has been greatly improved by revision. It is much easier to read, and its message is clearer. I have a few questions remaining about the methods and interpretation. I have also suggested a few grammatical, terminological, and clarification improvements directly on a pdf. For grammar, I highlighted the section of concern and used an inserted comment to suggest alternative wording. In your response, please reply explicitly to the points raised below. For the annotations on the pdf, you need only respond to those you do not agree with.

L146-149. Does Magurran (1988) justify the use of the Shannon Index as an estimate of niche breadth or does she just provide the formula? I checked Magurran's 2004 book, not having access to Magurran (1988). She doesn't mention Shannon Index as a measure of niche breadth and, in fact, notes that is not a reliable measure of diversity, although it does have a long history of use. Do you have more substantial references and/or a cogent explanation for why this measure and the evenness index are appropriate or accepted ways to measure what you need to know for this study? (I tried to check Platt et al. but it is a rather obscure journal and not available online through my university library.) Please note that I am not suggesting that you change the analysis if it is a reasonable measure but that you more clearly explain why this index is appropriate, at a minimum by citing references from well-established journals to indicate its established use.

L150ff, 186-188. Your overlapping group analysis also raises concerns. It seems likely to inflate sample size. I checked Wallace & Leslie for their justification, but they only comment on its heuristic value and do not justify its statistical validity, simply referring only to an unpublished Ph.D. from Zimbabwe. I don't see why a logistic regression using only one data point for each individual would not be more appropriate. I readily admit that I am not expert in statistics, so my concern is just that the references provided do not support this as an established, valid method. In addition, you should specify that you included all individuals in the sample. This would make it clear that increases in frequency of a category could be due to decreases in empty stomachs. Would it provide additional insight to examine size-related changes in empty guts (cases with no food remnants) and to see how the trends hold up when only individuals that provided food remnants are considered? Your conclusion that frugivory increases with size (L217) would not hold if what was really happening was that the proportion of cases with no food remnants was decreasing.

L183-184. When you refer to the patterns of niche breadth in your Results, you should be explicit whether you are referring to the Shannon or the Evenness Index or both. Also, your statements regarding patterns are not very robust as there is only a single measure for each category and no statistical support. I am willing to accept this as a simple, descriptive report, as similar statements occur in similar literature, but your contribution would be stronger if there were a way to develop confidence intervals.

Table 2. It is not clear why you have a row for 'unidentified material' without any values. Would it be useful to include the percent empty as well?

Fig. 1 could be improved in several ways. Reducing the unused extent of the x-axis (<1.9) would allow the figure to be larger in the final article. You need to provide the units and the log base for the CL measure. (For example, if a reader reasonably assumed that these were base 10 and cm, all your turtles would be > 1 m long, which I presume was not the case.) Even better would be to have arithmetic units on the x-axis even though the you still use a log scale. This approach is often used in publications to improve comprehension and is readily achieved in many graphics software. In this case, the caption would alert the reader to the log scale and use a subscript to indicate the base. You need to define the lines, which I assume are your regressions.

Fig. 3 represents Results and what this figure contributes should be included in the Results and properly cited.

---

## Round 0.3 · accepted · Accept

Thanks for your rapid response. The final changes were appropriately carried out, and I consider the manuscript now ready for publication.